# Self-assembly of a supramolecular hexagram and a supramolecular pentagram

Zhilong Jiang[1], Yiming Li[1], Ming Wang[2,3], Bo Song[3,4], Kun Wang[5], Mingyu Sun[5], Die Liu[1], Xiaohong Li[6], Jie Yuan[1], Mingzhao Chen[1], Yuan Guo[1], Xiaoyu Yang[1], Tong Zhang[5], Charles N. Moorefield[7], George R. Newkome[7], Bingqian Xu[5], Xiaopeng Li[3,4] & Pingshan Wang[1]

Five- and six-pointed star structures occur frequently in nature as flowers, snow-flakes, leaves and so on. These star-shaped patterns are also frequently used in both functional and artistic man-made architectures. Here following a stepwise synthesis and self-assembly approach, pentagonal and hexagonal metallosupramolecules possessing star-shaped motifs were prepared based on the careful design of metallo-organic ligands (MOLs). In the MOL design and preparation, robust ruthenium–terpyridyl complexes were employed to construct brominated metallo-organic intermediates, followed by a Suzuki coupling reaction to achieve the required ensemble. Ligand **LA** (**VRu$^{2+}$X**, V = bisterpyridine, X = tetraterpyridine, Ru = Ruthenium) was initially used for the self-assembly of an anticipated hexagram upon reaction with Cd$^{2+}$ or Fe$^{2+}$; however, unexpected pentagonal structures were formed, that is, **[Cd$_5$LA$_5$]$^{30+}$** and **[Fe$_5$LA$_5$]$^{30+}$**. In our redesign, **LB [V(Ru$^{2+}$X)$_2$]** was synthesized and treated with 60° V-shaped *bis*terpyridine (**V**) and Cd$^{2+}$ to create hexagonal hexagram **[Cd$_{12}$V$_3$LB$_3$]$^{36+}$** along with traces of the triangle **[Cd$_3$V$_3$]$^{6+}$**. Finally, a pure supramolecular hexagram **[Fe$_{12}$V$_3$LB$_3$]$^{36+}$** was successfully isolated in a high yield using Fe$^{2+}$ with a higher assembly temperature.

[1] Department of Organic and Polymer Chemistry, College of Chemistry and Chemical Engineering, Central South University, Changsha, Hunan 410083, China. [2] State Key Laboratory of Supramolecular Structure and Materials, College of Chemistry, Jilin University, Changchun, Jilin 130012, China. [3] Department of Chemistry and Biochemistry, Materials Science, Engineering, and Commercialization Program, Texas State University, San Marcos, Texas 78666, USA. [4] Department of Chemistry, University of South Florida, Tampa, Florida 33620, USA. [5] Single Molecule Study Laboratory, College of Engineering and Nanoscale Science and Engineering Center, University of Georgia, Athens, Georgia 30602, USA. [6] College of Chemistry, Chemical Engineering and Materials Science, Soochow University, Suzhou 215123, China. [7] Departments of Polymer Science and Chemistry, University of Akron, Akron, Ohio 44325, USA. Correspondence and requests for materials should be addressed to B.X. (email: bxu@engr.uga.edu) or to X.L. (email: xiaopengli1@usf.edu) or to P.W. (email: chemwps@csu.edu.cn).

Supramolecular self-assembly and recognition have become particularly appealing in recent years on scales ranging from relatively small and simple structures to complex, highly ordered architectures[1]. Garnering much of this attention, Lehn[2,3], Stang[4,5], Stoddart[6,7], Fujita[8–13], Raymond[14,15], Newkome[16–19], Leigh[20–25], Nitschke[26–29] and others[30–33] have designed and constructed numerous unprecedented metallo-architectures with precisely geometry controlled by using predesigned organic ligands through self-assembly procedures. Among the supramolecular assemblies with a variety of topology, mimicking the Star of David, a historical Hebrew symbol known as the Shield of David or Magen David, has proven to be a fascinating challenge in supramolecular chemistry arena. In one of Lehn's pioneering studies, tetra-, penta- and hexameric cyclic helicates were assembled by a linear organic $tris$(bipyridine) motif with $Fe^{2+}$ (refs 2,3). On the basis of hexameric circular helicates, Leigh et al. constructed an interlocking Star of David catenane through the connection of adjacent terminal organic bipyridine residues after self-assembly[20–22]. A similar strategy was also used to create the pentafoil knot[23] and Solomon's Knot[34], based on pentameric and tetrameric cyclic helicates, respectively.

In the field of terpyridine (tpy)-based supramolecular chemistry[35,36], precise control over the shape and size of the architectures is regarded as a tremendous challenge. Recent noteworthy accomplishments with tpy-based architectures include Sierpinski hexagonal gasket, spoked wheel, ring-in-ring and cage-like macromolecules[16–19,37–43]. Because of the ligand flexibility and deviation from the coordination geometry predicated by thermodynamic or kinetic stabilities[38] of several possible coordinated isomeric products, the assembly might produce a mixture and/or polymer, especially when targeting large and discrete assemblies (molecular weight > 10,000 Da). Therefore, appropriate ligand design and synthetic strategy play critical roles in the self-assembly process regarding the shape, size and complexity of the ultimate metallosupramolecule. To date, the multiple bipyridine-based one-step, pentagonal and hexagonal molecular assemblies have been reported in reasonable yields and could be shape-exchanged by using different counterions, such as $Cl^-$ and $SO_4^{2+}$ (refs 20–22).

The prospect of using octahedral $<tpy–M^{2+}–tpy>$ connectivity is considered to simplify the structural design and further, to introduce functionality for potential applications in sensing[44], magnetics[45], biomedicine[46,47], photonics[48,49] and catalysis[15,30,50,51].

Herein, we present the synthesis and self-assembly of planar supramolecular pentagram and hexagram structures, using precisely predesigned tpy-based metallo-organic ligands (MOLs).

## Results

The initial attempt to construct a supramolecular hexagram (Fig. 1) was performed by direct self-assembly of 1,2-$bis$terpyridine (V) and 1,2,4,5-tetra$kis$terpyridine (X) ligands, with metal ions in a precise stoichiometric ratio of 1:1:3 via one-pot procedure (Supplementary Figs 1 and 20–26). Unfortunately, direct self-assembly of V and X with $Cd^{2+}$ cations generated a thermodynamically stable small metallo-triangle $[Cd_3V_3]^{6+}$ and an unidentified metallo-polymer. Note that with $Cd^{2+}$, however, the resultant complex could not be separated by column chromatography because of its structural lability. Using $Fe^{2+}$ cations with strong coordination, we observed the formation of a mixture of small triangle, bowtie-like double-triangle and triple-triangle along with a small amount of unidentified metallo-polymer due to the self-sorting of polyterpyridines with metal ions (Fig. 2; Supplementary Information). With careful flash column chromatography, both small triangle $[Fe_3V_3]^{6+}$ and bowtie $[Fe_6V_4X]^{12+}$ were isolated and fully characterized by nuclear magnetic resonance (NMR) and electro-spray ionization mass spectrometry (ESI-MS; Supplementary Figs 3.1 and 4–16). A trace amount of triple-triangle $[Fe_9V_5X_2]^{18+}$ was only characterized with ESI-MS (Supplementary Figs 15 and 16) to confirm the proposed structure.

A stepwise strategy for constructing a MOL was considered to avoid the self-sorting of the same type of organic polyterpyridinyl ligands (Supplementary Fig. 2). In theory, bridging X and V with $Ru^{2+}$ would block the undesired assembly between the individual polyterpyridine ligands, for example, the dimer and trimer of V. Therefore, a stable $<tpy–Ru^{2+}–tpy>$ complex was

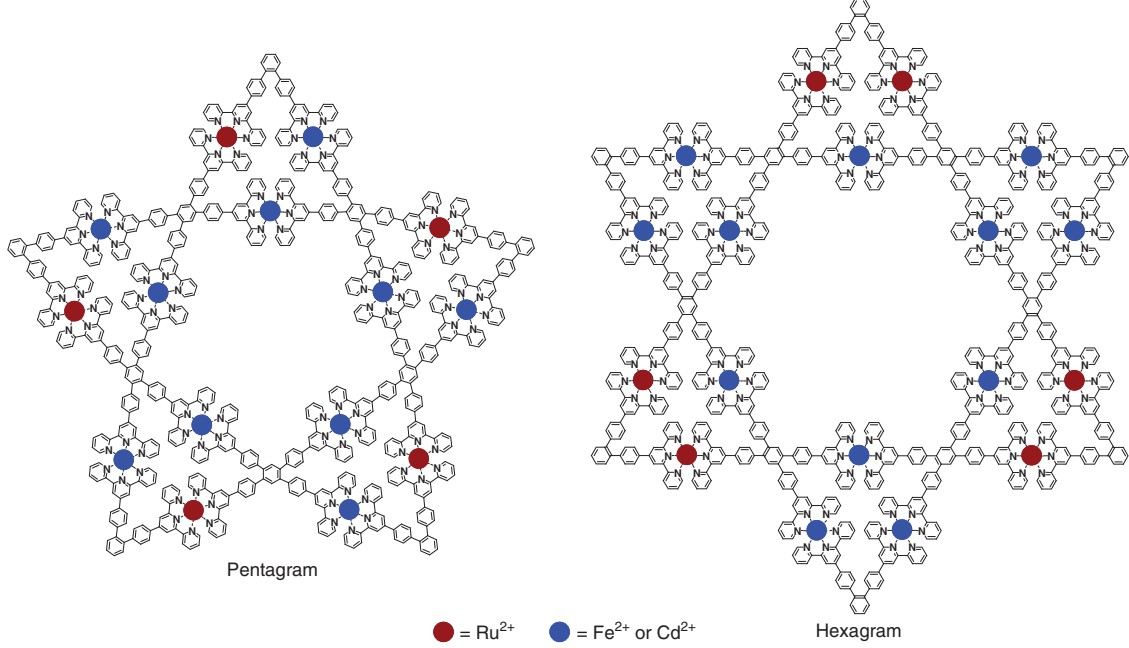

● = $Ru^{2+}$    ● = $Fe^{2+}$ or $Cd^{2+}$

**Figure 1 | Chemical structures of supramolecular pentagram and hexagram.** Star-shaped pentagram and hexagram contained five or six 1,2-bisterpyridines, five or six 1,2,4,5-tetraterpyridines and fifteen or eighteen metals, respectively.

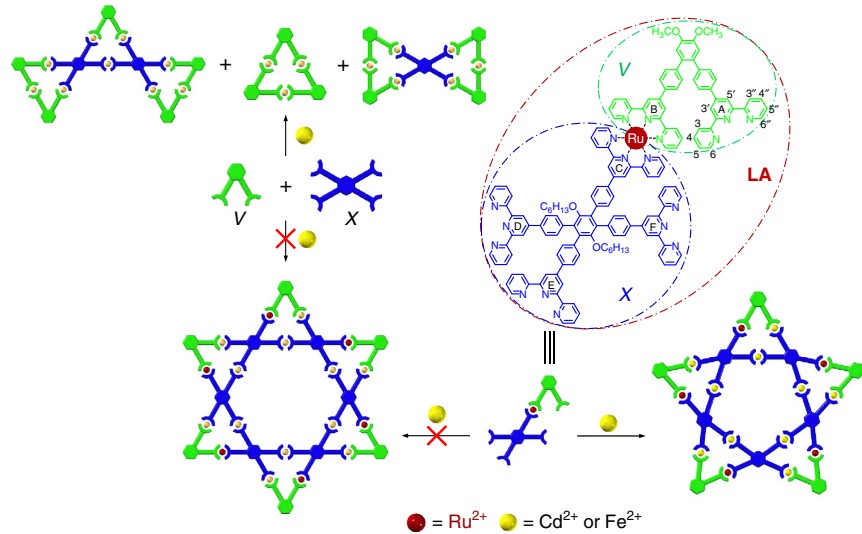

**Figure 2 | Illustration of an initial molecular hexagram designation.** Self-assembly of unexpected pentagram and multiple triangular assemblies in the initial design of hexagram.

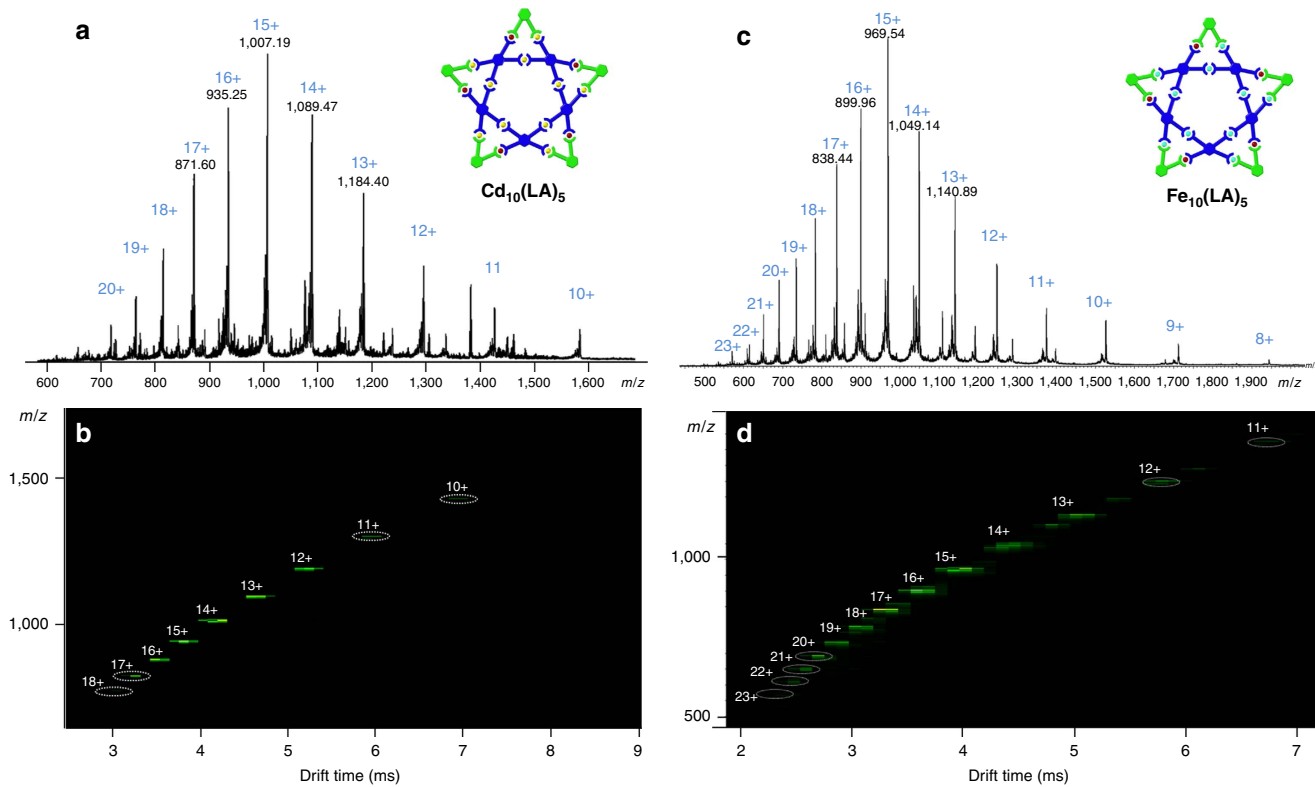

**Figure 3 | ESI/TWIM-MS spectrum.** (**a**) ESI-MS and (**b**) TWIM-MS of pentagonal **[Cd₁₀LA₅]³⁰⁺**; (**c**) ESI-MS and (**d**) TWIM-MS of pentagonal **[Fe₁₀LA₅]³⁰⁺**.

chosen as the connector to bridge **X** and **V** as MOL for the purpose of creating the desired star-shaped pattern. The supramolecular Star of David was expected to create upon coordination of **LA** with $Fe^{2+}$ or $Cd^{2+}$ cations based on the advantage of high degree of reversible coordination, consequently leading to the desired architecture. This stepwise strategy would prevent the formation of triangle, for example, $[Cd_3V_3]^{6+}$, from the self-sorting coordination of **V** with $Cd^{2+}$ and metallo-polymers from multiple complexations. In this approach, the self-assembly could be forced into desired six star-shaped motifs. When direct mixing $Ru^{2+}$ with **V** and **X** (that is, $V + RuCl_3 +$

$X \neq [VRuX]^{2+}$, or **LA**) only generated a complex mixture (Supplementary Methods) in which the isolation and purification were particularly challenging due to their significant polarity and poor solubility. Subsequently, a coupling on the complex strategy was applied to construct **LA** based on the remarkable stability of the $<tpy–Ru^{2+}–tpy>$ connector. As shown in Supplementary Fig. 2, a monobromoterpyridineruthenium trichloride adduct was attached to one tpy of **X** to generate a key intermediate, **L-Br** (Supplementary Figs 17 and 27–32). The Suzuki cross-coupling of **L-Br** with tpy–$C_6H_4$–B(OH)₂ introduced another free tpy to form **LA** (Supplementary Figs 18 and 33–36), with

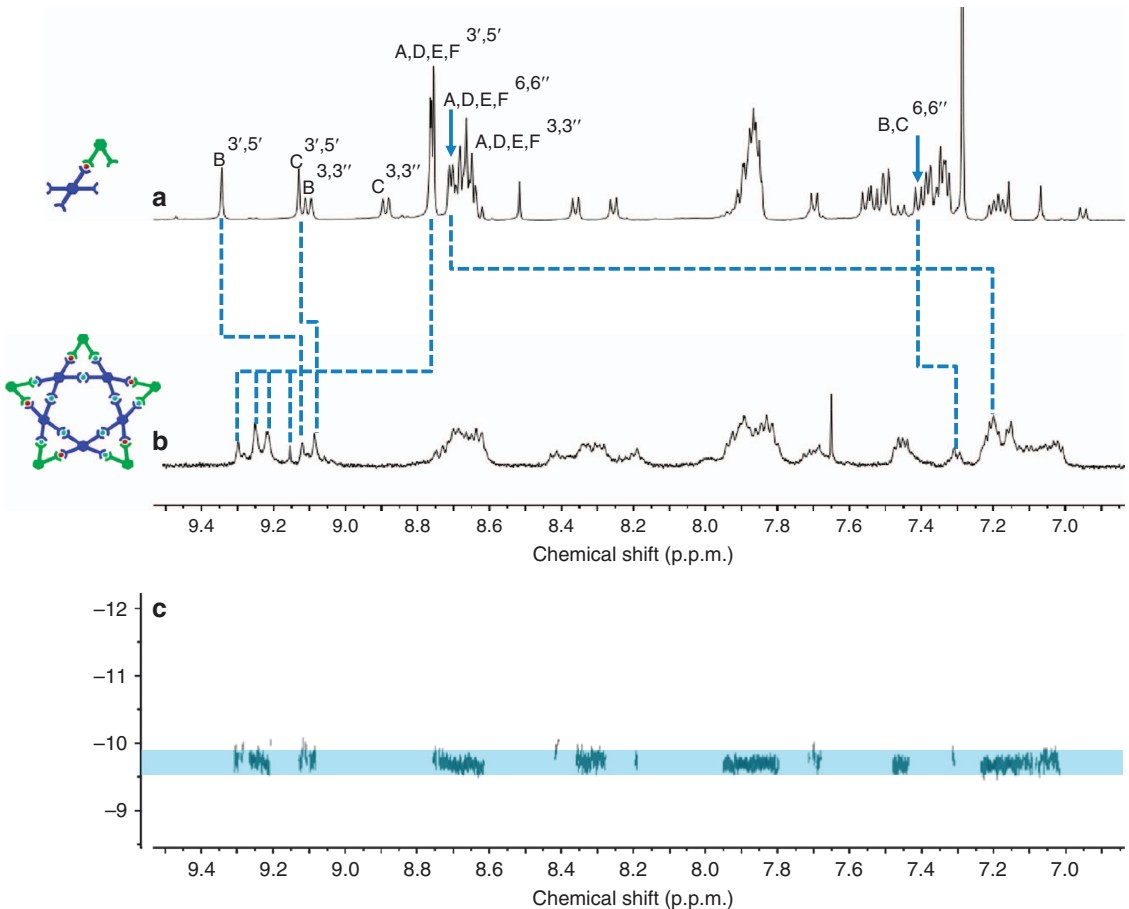

**Figure 4 | $^1$H NMR spectrum.** (**a**) **LA** and (**b**) molecular pentagram **[Fe$_{10}$LA$_5$)]**; (**c**) DOSY of **[Fe$_{10}$LA$_5$)]** (500 MHz, 25 °C, CDCl$_3$ for ligands and CD$_3$CN for supramolecule).

$<$tpy–Ru$^{2+}$–tpy$>$ connector (45%) after column chromatography (Al$_2$O$_3$, CHCl$_3$/MeOH).

The asymmetric metallo-ligand **LA** containing four uncomplexed tpy moieties was mixed with Cd$^{2+}$ or Fe$^{2+}$ in 1:2 ratio (Supplementary Methods), respectively. After counterions exchange using excess NH$_4$PF$_6$, the resultant reddish powder was obtained in nearly quantitative yield ($>$95%). From geometrical and topological points of view, the one-step assembly of supramolecular building block **LA** with Cd$^{2+}$ or Fe$^{2+}$ should result in the highly ordered, six-pointed hexagram (Fig. 2). Surprisingly, the assembled products were shown to be the pure pentagrams with composition of **[Cd$_{10}$LA$_5$]$^{30+}$·30PF$_6^-$** and **[Fe$_{10}$LA$_5$]$^{30+}$·30PF$_6^-$**, respectively, after careful analysis of its high-resolution ESI-MS data (Fig. 3a,c). Therefore, the selectivity of this monomer **LA** favoured five-membered supramolecular structures. The mass spectrum of **[Cd$_{10}$LA$_5$]$^{30+}$** revealed a series of peaks with charge states from 10+ to 18+ derived by the successive loss numbers of PF$_6^-$ counterions. Similarly, **[Fe$_{10}$LA$_5$]$^{30+}$** displayed similar peaks with charge states from 11+ to 23+. Both were analysed on the basis of the mass-to-charge ratios of molecular weights of 17,282 and 16,716 Da for **[Cd$_{10}$LA$_5$]$^{30+}$** and **[Fe$_{10}$LA$_5$]$^{30+}$**, respectively. The isotope patterns for each charge state were in excellent agreement with the corresponding simulated isotope peaks (Supplementary Figs 48–51). Furthermore, the ESI-travelling wave ion mobility-mass spectrometry (ESI-TWIM-MS)[37] data (Fig. 3b,d) supported the pentagonal structural assignment by exhibiting a single band with a narrow drift time distribution for each charge state, indicating that no other structural conformers, isomers or other components were detected. The experimental collision cross-

section calculated from ion mobility was also compared to theoretical CCS obtained from molecular modelling for additional support[52,53]. The experimental CCS of each charge state of **[Cd$_{10}$LA$_5$]$^{30+}$** and **[Fe$_{10}$LA$_5$]$^{30+}$** gave an average CCS at 2,369 and 2,345 Å$^2$, which are slightly smaller than the average theoretical CCS of one hundred energy-minimized modelling structures at 2,458 and 2,422 Å$^2$, respectively.

The $^1$H NMR spectra of **[Fe$_{10}$LA$_5$]$^{30+}$** in Fig. 4b and **[Cd$_{10}$LA$_5$]$^{30+}$** (Supplementary Figs 41–44) were complicated due to multiple overlapping polyterpyridine environments. In the $^1$H NMR spectrum of **LA** (Fig. 4a), two singlets of **[Fe$_{10}$LA$_5$]$^{30+}$** $\sim$9.25 and 9.30 p.p.m. were assigned to the tpy$H^{3',5'}$ protons for the $<$tpy–Ru$^{2+}$–tpy$>$ moieties, which only showed the small shifts; the tpy$H^{3',5'}$ of newly formed $<$tpy–Fe$^{2+}$–tpy$>$ linkages exhibited the expected large downfield shifts, as a result of the lower electron density upon complexation. Furthermore, the upfield shift in the tpy$H^{6,6''}$ protons is attributed to the shielding effect of the metal centres. These observed chemical shifts and ultraviolet–visible (UV–vis) measurements (Supplementary Fig. 56) indicated the expected $<$tpy–Fe$^{2+}$–tpy$>$ linkages were formed. UV–vis spectroscopies of **[Cd$_{10}$LA$_5$]$^{30+}$**, **[Fe$_{10}$LA$_5$]$^{30+}$** and **[Fe$_{12}$V$_3$(LB)$_3$]$^{36+}$** in CH$_3$CN (for PF$_6^-$ as the anions) showed the expected absorbance patterns at *ca.* 495 and 575 nm for the tpy–Ru$^{2+}$–tpy and tpy–Fe$^{2+}$–tpy units, respectively. Moreover, six signals of tpy$H^{3',5'}$ were consistent with the desired structure. Peak assignments were confirmed using two-dimensional 2D-NOESY NMR spectra (Supplementary Figure 47) NMR spectra. The diffusion-ordered NMR spectroscopy (DOSY) was further applied to confirm the structure, which showed **[Fe$_{10}$LA$_5$]$^{30+}$** as a single component with the diffusion

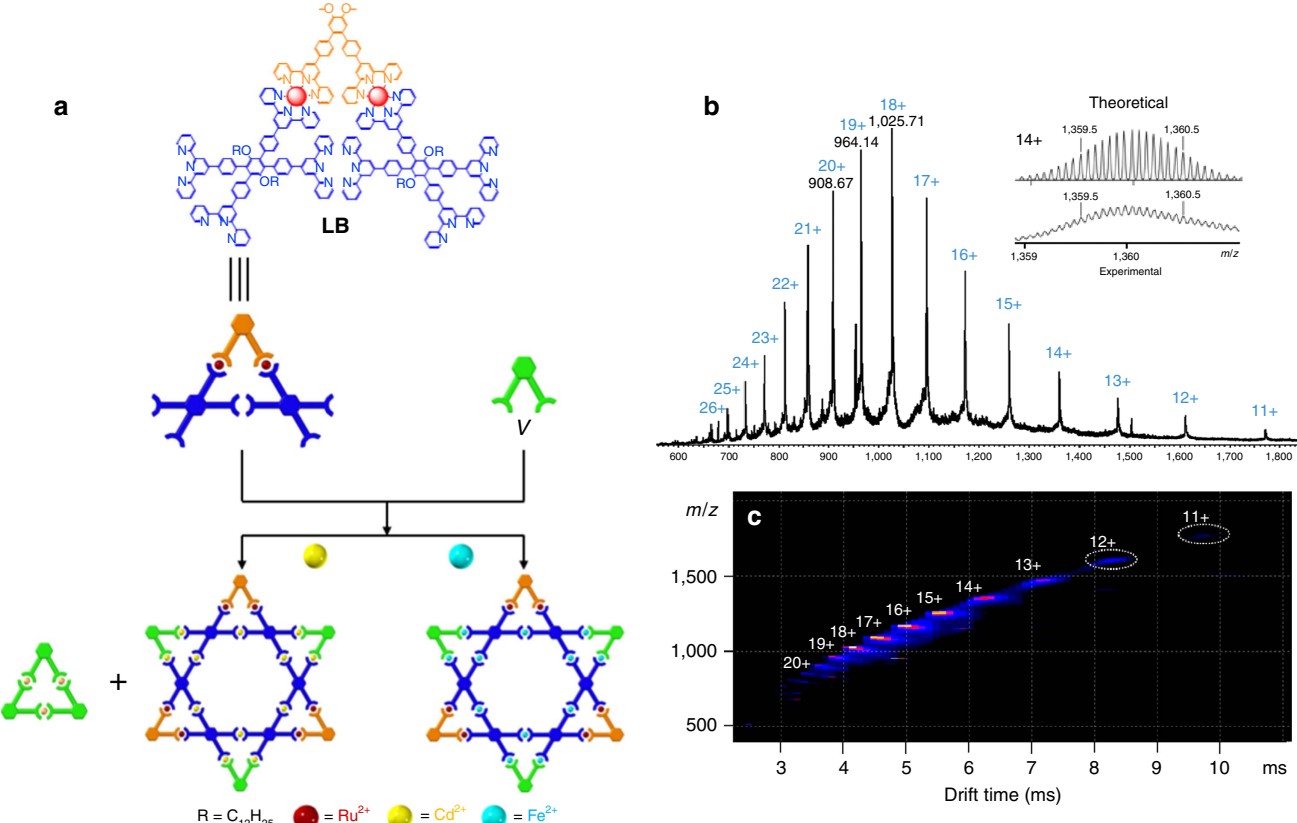

**Figure 5 | Self-assembly of supramolecular hexagram [Fe₁₂V₃LB₃]³⁶⁺.** (**a**) Schematic illustration of synthesizing molecular Star of David by using metallo-ligand **LB**, **V** and metals. (**b**) ESI-MS and (**c**) TWIM-MS of metallo-hexagram.

coefficients of $1.99 \times 10^{-10} \, \mathrm{m^2 \, s^{-1}}$ at 298 K (Fig. 4c). The experimental hydrodynamic radius ($r_H = 2.99$ nm) was calculated and it agrees well with the diameter of molecular modelling.

The formation of pentagram instead of hexagram is most likely attributed to the slight flexibility of the extended terpyridines and deviation from typical coordination with the metal ions[54]. The different side lengths of each individual dinuclear triangle in the pentagram may play a vital role in determination of the formation of the five- versus six-pointed star's supramolecular architecture. **LA** reacted as an asymmetrical piece in which the edge lengths (<tpy–Ru–tpy>, <tpy–Cd–tpy> or <tpy–Fe–tpy>) in each identical triangle (Fig. 2) should have a slight variation. This is confirmed through the molecular modelling or reported crystal structures (18.004, 18.672 and 17.821 Å for <tpy–Ru²⁺–tpy>, <tpy–Cd²⁺–tpy> and <tpy–Fe²⁺–tpy>, respectively; Supplementary Fig. 57). Nevertheless, this subtle difference of lengths between <tpy–Ru²⁺–tpy> and <tpy–Fe²⁺–tpy> or <tpy–Cd²⁺–tpy> could generate different structural outcomes.

To assemble a six-pointed Star of David architecture, an alternative synthetic strategy was developed based on a new metallo-ligand (**LB**; Fig. 5a; Supplementary Figs 3,19 and 37–40). When a *bis*tpy-RuCl₃ adduct, **S5**, coordinated with 3.0 equiv. of *tetra*terpyridine **X**, a symmetric metallo-ligand **LB** was isolated via column chromatography (Al₂O₃, eluting with CHCl₃/MeOH) in *ca.* 31% yield. The first attempt at assembling the supramolecular Star of David was by directly mixing **LB** and **V** with Cd²⁺ in a precise 1:1:4 stoichiometric ratio under conventional conditions. After counterion exchange with PF₆⁻, a pale red precipitate was formed and then washed with deionized water and MeOH to afford (90%) a reddish product. Unfortunately, ¹H NMR and DOSY measurements indicated the presence of two distinct products (Supplementary Fig. 45). In addition, this

mixture was analysed by high-resolution ESI-MS (Supplementary Figs 52 and 53), which revealed the major assembled product was the desired metallo-hexagram with molecular composition of [Cd₁₂V₃LB₃]³⁶⁺·36PF₆⁻ and along with a small amount of metallo-triangle [Cd₃V₃]⁶⁺·6PF₆⁻ derived from the assembly of **V** with Cd²⁺. Although the self-assembly of **LB**, **V** with Cd²⁺ had been optimized to avoid the metallo-triangular formation of self-sorting in terms of different temperatures, solvents, counter-ions and starting material ratio, it was difficult to prevent the formation of small triangle. Moreover, we were not able to remove the small triangle from the Star of David mixture through conventional column chromatography because of the weak coordination of <tpy–Cd²⁺–tpy> during separation process.

Considering the greater stability of <tpy–Fe²⁺–tpy> coordination comparison to <tpy–Cd²⁺–tpy>, the combination of **LB**, **V** and Fe²⁺ in a stoichiometric ratio of 1:1:4 at a higher temperature condition (~140 °C) in an ethylene glycol solution overnight was performed. Following the processes of the anion exchange with PF₆⁻, rinse with distilled water and MeOH, dry *in vacuo* for 12 h; the product was left to yielding a reddish powder. With a short-path column chromatography (MeCN/H₂O/NaNO₃), the metallo-hexagram [Fe₁₂V₃LB₃] was successfully isolated as the major product, along with a small fraction (~5%) of metallo-triangle [Fe₃V₃], which is identical to the one isolated in our initial self-assembly attempt (Fig. 2). The high-resolution ESI-MS spectrum (Fig. 5b; Supplementary Figs 54 and 55) demonstrated a series of adequate peaks with charge states from 11+ to 25+ identified by the loss of PF₆⁻ counterions. All of the peaks' isotopic patterns corresponded to the calculated isotopic distributions of metallo-pentagram, [Fe₁₂V₃LB₃]³⁶⁺·36PF₆⁻ (molecular weight = 21,069 Da). In ESI-TWIM-MS (Fig. 5c), a series of signals with narrow drift times were observed, indicating

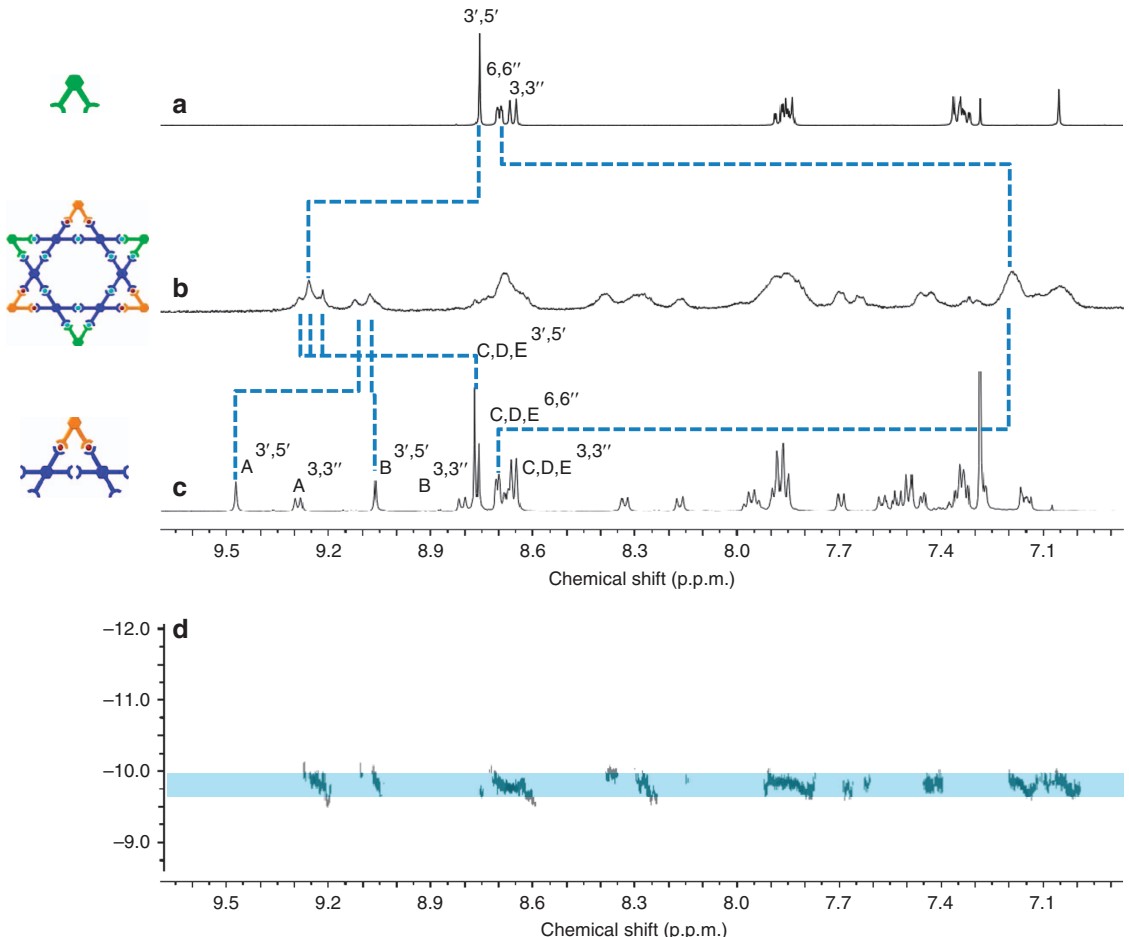

**Figure 6 | Comparison of $^1H$ NMR spectrum.** (**a**) *bis*terpyridine ligand (500 MHz, 25 °C, CDCl₃), (**b**) molecular Star of David **[Fe₁₂V₃LB₃]** (500 MHz, 25 °C, CD₃CN), (**c**) metallo-ligand **LB** (500 MHz, 25 °C, CD₃CN) and (**d**) DOSY of **[Fe₁₂V₃LB₃]** (500 MHz, 25 °C, CD₃CN).

that a discrete and rigid molecular metallo-pentagram was assembled. The average experimental CCS was calculated as 2,843 Å², which is close to the average theoretical CCS at 2,980 Å² from molecular modelling.

## Discussion

The $^1H$ NMR spectra revealed the broad peaks for polyaromatic regions (Supplementary Figs 46 and 47); however, the specific chemical shifts upon complexation could be distinguished when compared with both precursors, that is, **V** and **LB**. The $^1H$ NMR spectrum of molecular metallo-pentagram showed two sets of characteristic shifts for tpy$H^{3',5'}$. The first set of peaks is for the coordination of <tpy–$Fe^{2+}$–tpy>, which shows a downfield shift to *ca.* 9.25 p.p.m. The second set was assigned for the original <tpy–$Ru^{2+}$–tpy> at *ca.* 9.1 p.p.m. The proton of uncomplexed free tpy$H^{6,6''}$ in **V** and **LB** shifted upfield to 7.2 p.p.m. from *ca.* 8.7 p.p.m. after self-assembly due to the electron shielding effects. In the nonaromatic region, there is only one set of signals, which can be assigned to the –OMe moieties at *ca.* 4.0 p.p.m. and alkyoxy chain for –O$CH_2$– at 3.5 p.p.m. The full assignment of ligands and assembled architectures were verified and confirmed by 2D-COSY and 2D-NOESY experiments. DOSY unambiguously showed **[Fe₁₂V₃LB₃]** as a single component with the diffusion coefficients of $1.58 \times 10^{-10}$ m² s$^{-1}$ at 298 K (Fig. 6d). The experimental hydrodynamic radius ($r_H = 3.77$ nm) was approximately consistent with the theoretical molecular diameter.

The anions of $PF_6^-$, $Cl^-$, $NO_3^-$ and $BF_4^-$ had been used to investigate the structural influence during the anion exchanges by

means of high-resolution ESI-MS measurements. We found that there is no molecular structure transformation when utilizing different anions as the counterions (Supplementary Fig. 59), the structure is stable even in a mixed counterion systems. It may due to the bigger molecular central holes (>3.5 nm) and larger molecular weight (close to 20,000 Da)[36].

Furthermore, a droplet of hexagram (Fig. 7) or pentagram (Supplementary Fig. 58) solutions in acetonitrile (~$10^{-7}$ M) were deposited on the surface of newly cleaved mica or highly ordered pyrolytic graphite (HOPG), for atomic force microscopy (AFM) or room temperature scanning tunnelling microscopy (STM). The AFM image showed individual hexagonal shape and central hole (Fig. 7a,b). In the STM imaging, hexagram supramolecules were assembled into ribbon-like pattern on HOPG (Fig. 7d). The individual molecule (Fig. 7c) was also distinguished with more details than the AFM images. However, the exact shape of the molecules has a little distortion, which is common in room temperature STM imaging.

In summary, we have constructed a supramolecular pentagram and a supramolecular hexagram using metal–organic building blocks with strong <tpy–$Ru^{2+}$–tpy> connectivities through stepwise strategies. Introducing Ru-polyterpyridyl moieties in the self-assembly with the weak coordination metal ions, that is, $Cd^{2+}$ or $Fe^{2+}$, successfully blocked the self-sorting of individual organics, and thus achieved the self-assembly of giant discrete star-shaped metallo-architectures. The fractal-like pentagonal and hexagonal architectures were fully characterized by high-resolution ESI-MS, TWIM-MS, NMR, 2D-NOESY and DOSY

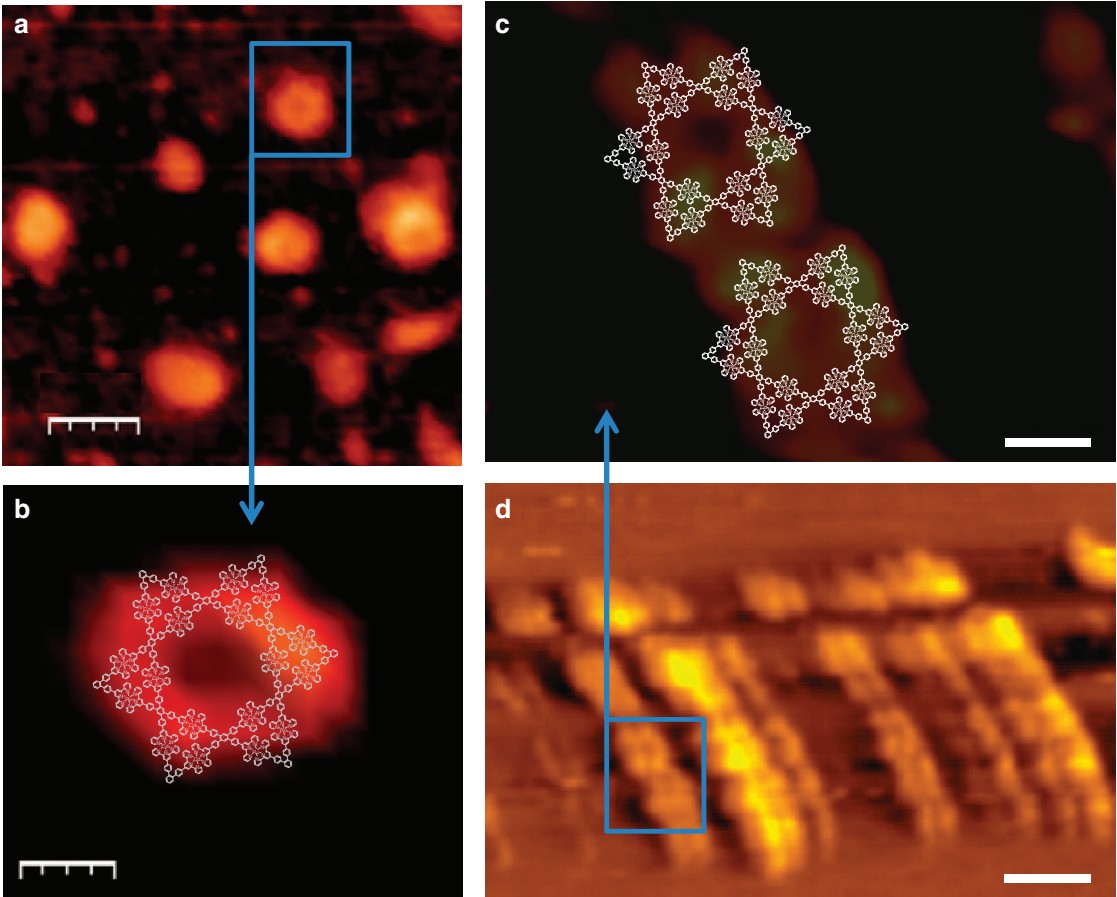

**Figure 7 | AFM and STM images of metallo-hexagram.** (**a**,**b**) AFM images on mica (scale bar, 40 and 9.3 nm, respectively). (**c**,**d**) STM images on HOPG at ambient condition (scale bar, 5 and 25 nm, respectively).

spectroscopies as well AFM and STM measurements. Using well-established coordination-mediated programmable assemblies and stepwise strategies, we may pave a new avenue towards a new series of ruthenium-based multi-nuclear metallosupramolecules with precisely controlled architecture, increasing complexity and ultimately tailored functionality.

## Methods

**Sample preparation.** All starting materials were purchased from Aldrich and Alfa Aesar, and were used without further purification. Complex $[Fe_3V_3]^{6+}$ is consistent with the results published by Newkome[36]. Column chromatography was conducted by using basic $Al_2O_3$ (sinopharm chemical reagents co., Ltd, 200–300 mesh) or $SiO_2$ (Qingdao Haiyang Chemical co., Ltd, 200–300 mesh). The $^1H$ NMR and $^{13}C$ NMR spectra were recorded on a Bruker Avance 400-, 500- and 600-MHz NMR spectrometer in $CDCl_3$, DMSO-$D_6$ and $CD_3CN$ with tetramethylsilane (TMS) as the inner standard. UV–vis absorption spectra were recorded with an Agilent 8453 UV–vis Spectrometer. Photoluminescence spectra were recorded on a Hitachi 2500 Luminescence spectrometer. ESI mass spectra were recorded with a Waters Synapt G2 tandem mass spectrometer, using solutions of 0.01-mg sample in 1 ml of $CHCl_3/CH_3OH$ (1:3, v/v) for ligand or 0.5 mg in 1 ml of MeCN, MeOH or MeCN/MeOH (3:1, v/v) for complex.

**TWIM-MS.** TWIM MS experiments were performed under the following conditions: ESI capillary voltage, 3 kV; sample cone voltage, 30 V; extraction cone voltage, 3.5 V; source temperature 100 °C; desolvation temperature, 100 °C; cone gas flow, 101 h$^{-1}$; desolvation gas flow, 7001 h$^{-1}$ ($N_2$); source gas control, 0 ml min$^{-1}$; trap gas control, 2 ml min$^{-1}$; helium cell gas control, 100 ml min$^{-1}$; ion mobility (IM) cell gas control, 30 ml min$^{-1}$; sample flow rate, 5 μl min$^{-1}$; IM travelling wave height, 25 V; and IM travelling wave velocity, 1,000 m s$^{-1}$. Q was set in rf-only mode to transmit all ions produced by ESI into the tri-wave region for the acquisition of TWIM-MS data.

**Molecular modelling.** Energy minimization of the macrocycles was conducted with the Materials Studio version 6.0 program, using Anneal and Geometry Optimization tasks in the Materials Studio Forcite module (Accelrys, Inc.).

**Microscopy analysis.** Transmission electron microscopy was obtained on JEOL 2010. STM: the sample was dissolved in DMF or $CH_3CN$ at a concentration of 5.0 mg ml$^{-1}$. Solution (5 μl) was dropped on HOPG surface. After 30 s, surface was washed slightly with water for three times and totally dried in room temperature in air. The STM images were taken with a PicoPlus SPM system using a PicoScan 3000 Controller. The obtained STM images were processed by WSxM software. AFM: the 40 N m$^{-1}$ rectangle AFM tip was used to make markers on the surface of the sample with nanoshaving technique. The size of the markers is $10 \times 10$ μm and the distances between two closest markers are 30 μm. The loading forces used are 6 μN and shaving speed is ∼5 μm s$^{-1}$.

**Data availability.** The data that support the findings of this study are available from the corresponding author on request.

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

## Acknowledgements

This research was supported by the National Natural Science Foundation of China (21274165 for P.W.; 21305098 for X.L.), the Distinguished Professor Research Fund from Central South University of China (for P.W.), the Fundamental Research Funds for the Central Universities from Central South University (2013zzts014 for P.W.), NSF (CHE-1506722 and DMR-1205670 for X.L.; ECCS-1609788 for B.X.; CHE-1151991 for G.R.N.), the Research Corporation for Science Advancement (23224 for X.L.) and the ACS Petroleum Research Fund (55013-UNI3 for X.L.). We gratefully acknowledge Dr Carol D. Shreiner for her professional consultation. We also acknowledge the NMR measurements from The Modern Analysis and Testing Center of CSU.

## Author contributions

All authors have given approval to the final version of the manuscript. Z.J., X.L. and P.W. designed the experiments. Z.J., M.W. and J.Y. completed the synthesis. Z.J., Y.L., M.W., B.S., K.W., X.L., M.S., B.X., D.L., M.C., Y.G., T.Z. and X.Y. carried out the characterization studies. Z.J., Y.L., C.N.M., G.R.N., B.X., X.L. and P.W. analysed the data and wrote the manuscript. All the authors discussed the results and commented on and proofread the manuscript.
