## [Peer Review File · Nature Communications]

Reviewers' comments:

Reviewer #1 (Remarks to the Author):

In this manuscript the authors report the synthesis and characterization of one 15 metal-ion-containing coordination complex pentagram they call a 'Star-of-Texas' and an 18 metal-ion-containing coordination complex hexagram they call a 'Star-of-David'.

The work is a significant achievement in the domain of terpyridine coordination complexes with iron, cadmium and ruthenium. On the whole the manuscript is written reasonably clearly. However, some sentences are incomplete or muddled and there are a number of typographical and other errors.

The lack of any X-ray characterized structures in the domain of giant multi nuclear complexes is a problem for demonstrating that the claimed architecture has been obtained. The poor quality and complex NMR spectra do not help either, but the authors manage to support their results with various techniques which are ultimately convincing.

The following points need addressing:

1. The choice of the name 'Star of David' and, in particular, 'Star of Texas' for these structures is not, in my opinion, particularly appropriate nor useful. Star of David has some justification as the molecular structure can reasonably be seen as a 2D representation of the Star of David. However the Star of Texas (or Lone Star) is a particular type of five-pointed star possessing either 3D-bevelling or no internal lines, and so has a different topology to the synthesized molecule, the only similarity being that they are (different types of) five-pointed stars. Maybe the authors should simply call their structures 'supramolecular hexagram and pentagram' which is exactly what they are.

2. In line 61 hexameric should be tetrameric.

3. In lines 75-76 the authors state that "To date, few one-step, Star of David molecular assemblies of (?) have been reported in reasonable yields due, in part, to its large hollow, non-compact scaffold" but no references are included to support this. Alexandropoulos et al.'s work (Chem. Comm. 2016, 52, 1693) should be referenced and the authors' rationale as to why these assemblies are rarely reported is just unsupported, somewhat random, speculation.

4. In line 104 the authors suggest that Zn²⁺ could be used in the self-assembly reactions but there is no further mention of this. It would be better not to mention it at all if no experiments were actually performed using this metal.

5. Throughout the manuscript and ESI, metal sources are represented as Mn⁺. It would be clearer to specify the full metal salt used, e.g. FeSO₄·7H₂O, Cd(NO₃)₂·6H₂O, especially in schemes. The counter-ion for charged species is not stated in their assignment section, e.g. the 'Five star of Star-of-Texas' is referred to as [Fe₁₀(LA)₅]³⁰⁺ in section 2.8 of the ESI. Similarly, counter ions are not stated in figures of the NMR spectra for charged species, e.g. in figure S41 the species is stated to be [Fe₁₀(LA)₅]³⁰⁺. As counter ions can influence NMR spectra of charged species they should be stated.

6. The complex notation "<tpy-Mn⁺-tpy>" could be represented in a neater way. Perhaps simply [M(tpy)₂]ⁿ⁺ or [M(tpy)(tpy')]ⁿ⁺.

7. In the Results and Discussion section (line 98) the authors claim that the [M₃V₃] complexes are "enthalpically favoured" but offer no evidence that this is the case. Surely there is no enthalpic advantage to forming this triangle over the hexa- or nona-metallic species, only an entropic

advantage.

8. In lines 169-171, the authors compare the potential stabilities of self-assembled constructs to five- and six-membered rings. This comparison appears tenuous and a little confusing and it does not have any relevance to the observations.

9. Though the use of computer generated "theoretical" molecular diameters is fine for estimating the sizes of the reported constructs, some more detailed explanation on software/techniques used etc. should be provided in the ESI.

10. The captions for Figures 4 and 6 should contain solvent/temperature/magnet strength for NMR data.

11. The caption for Figure 5 says "Synthesis of key metallo-ligand LB and hexagonal supramolecular Star-of-David assemblies" but only the self-assembly steps are shown. The caption needs altering to reflect this.

12. The TEM images do not appear to support the given data. For example, Figure 6E shows dots of radius approx. 3.3 nm if the scale bar is correct, rather than the reported 7.3 nm. There is a similar discrepancy in Figure 6F.

13. In the ESI, scheme S2 is not correct: the methoxy groups are in ortho and meta and not para and meta. Molecule S2 should be on the arrow for synthesis of LA from L-Br.

14. The procedure for synthesis of complex S4 does not have a yield.

15. The procedure for synthesis of complex L-Br states that 150.6 g rather than 150.6 mg of X was used.

16. All figures of NMR spectra in the ESI should include the solvent used in the caption.

17. The ^1H NMR spectrum in figure S41 has a high signal: noise ratio. Hence the integrations shown do not match well with those stated for the assignment of $[\text{Fe}_{10}(\text{LA})_5]^{30+}$. This is also the case for $[\text{Fe}_{12}\text{V}_3(\text{LB})_3]^{36+}$.

18. Some NMR spectra are reported to have been recorded operating at a ^1H resonance frequency of 600 MHz (e.g. Figure S8). This spectrometer is not recorded in the General Procedures section.

19. In section 2.9 the ESI-MS peak at $m/z = 942.4$ is incorrectly assigned as the 19+ rather than the 20+ species.

20. The ^1H NMR spectrum (figure S44) of $[\text{Fe}_{12}\text{V}_3(\text{LB})_3]^{36+}$ has not been assigned (section 2.10). The quality of the spectrum is similar to figure S41, which has been assigned (section 2.8).

21. The caption of figure S44 is highlighted and the species should read $[\text{Fe}_{12}\text{V}_3(\text{LB})_3]^{36+}$ instead of $[\text{Fe}_{12}\text{V}_3(\text{LB})_3]^{36+}$.

Reviewer #2 (Remarks to the Author):

In supramolecular chemistry arena, the construction of Star-of-Texas and Star-of-David has proven to be a fascinating challenge. In this manuscript, through a stepwise synthetic strategy and self-assembly approach, Li and Wang et al. carefully designed and prepared two new pentagonal and hexagonal metallosupramolecules possessing the Star-of-Texas and Star-of-David motifs. The

Star-of-Texas and Star-of-David architectures were fully characterized by high resolution ESI-MS, TWIN-MS, NMR, 2D-NOESY, and DOSY spectroscopies as well TEM. This is an interesting work that I would recommend to be published in Nature Communication after minor revision.

1. During the past years, Lehn and Leigh groups have also successfully constructed a series of Star-of-Texas and Star-of-David architectures through the coordination-driven self-assembly (J. Am. Chem. Soc., 1997, 119, 10956; Nat Chem., 2014, 6, 978-982; Nat. Chem., 2012, 4, 15-20; etc). The authors need to discuss the advantages of their method compared to the methods reported by Lehn and Leigh to prepare Star-of-Texas and Star-of-David architectures;
2. For preparing the Star-of-Texas and Star-of-David architectures, Cd(NO₃)₂ and FeCl₂ have been used in this work. I wonder what will happen if CdCl₂ and Fe(NO₃)₂ are used.
3. In order to prepare the Star-of-Texas architecture via one-pot procedure, ligand V and ligand X were mixed directly with FeSO₄, Unfortunately, a mixture of small triangle, bowtie-like double-triangle, and triple-triangle along with a small amount of unidentified metallo-polymer was formed. Considering the successful preparation of Star-of-Texas architecture by using FeCl₂, I also wonder what will happen if FeCl₂ is used for preparing the Star-of-Texas architecture via one-pot procedure.
4. In page 7, the authors described that "These observed chemical shifts and UV-vis measurements (see ESI, Figure S54) indicated the expected <tpy-Fe²⁺-tpy> linkages were formed". The detailed discussion of the UV-vis results should be provided. Especially the relationship between the UV-vis spectra and the architecture structure should be discussed in detail.
5. In the past few years, the author's group has prepared a series of metallocupramolecular architectures with fascinating structure based on terpyridine (tpy) skeleton (for example, J. Am. Chem. Soc., 2014, 136, 6664-6671; J. Am. Chem. Soc., 2014, 136, 10499-10507; J. Am. Chem. Soc., 2015, 137, 1556-1564; Chem. Commun., 2016, 52, 9773; etc.). Although the construction of such beautiful metallocupramolecular architectures is interesting, it would be better if some applications will be explored for these architectures.
6. Some errors should be corrected, for instance, ml should be corrected as mL; a space should be added between the value and the unit.

Reviewer #3 (Remarks to the Author):

The article by Jiang et al is a very thorough piece of work. The authors report the systematic synthesis of coordination assemblies based on Star-of-Texas and Star-of-David arrangements. The work is undoubtedly clever and the strategy interesting but I am left wondering whether it is a significant advance beyond previous work?

The authors identify a large number of previous papers on related materials and even acknowledge the previous study of a Star-of-David catenane (ref 20) which in a similar fashion to the current study builds upon the circular helicates reported Lehn twenty years ago (references 2 and 3). Is this current study a significant conceptual advance on these previous studies? Ultimately I don't think so and therefore I do not see that the paper is sufficiently novel for publication in Nature Communications.

Bearing in mind my reservations about the novelty of this work I do have some questions about the work as reported. Lehn's original work on circular helicates identified the role of the anion as significant in determining the nature of the structure of the resulting coordination assembly. What is the role of the anion in the current study? Is the size and shape of the PF₆⁻ anion significant in determining the target Star-of-Texas or Star-of-David arrangements? Can the balance between different arrangements be modified by different anions. Additionally is the anion trapped within the structure as is the case for Lehn's circular helicates? Have the authors recorded ¹⁹F NMR which would help to identify the role of the anion?

The authors need to seriously consider these additional studies and clarify if there are additional factors which control the assembly of their structures.

Reviewer #4 (Remarks to the Author):

This is a potentially interesting ms. However, the write up is very confusing, cumbersome and difficult to follow. There are also numerous English problems. I doubt that coauthor Newcome saw this ms and if he did he should never have let it be submitted with such poor English and write up.

However, there are even more serious issues. Mass spec does not prove structure at best it can only give stoichiometry. That leaves just proton NMR for characterization and frankly that is simply not sufficient for complex molecules like these. Just as an example the H-NMR of both compounds are essentially the same and makes it impossible to really assign structure. Hence until there is better and more definitive structure proof and a much better and clear write up this ms is NOT ready for publication in any journal let alone a Nature Journal.

Response to Reviewers' comments:

Reviewer #1 (Remarks to the Author):

In this manuscript the authors report the synthesis and characterization of one 15 metal-ion-containing coordination complex pentagram they call a 'Star-of-Texas' and an 18 metal-ion-containing coordination complex hexagram they call a 'Star-of-David'.

The work is a significant achievement in the domain of terpyridine coordination complexes with iron, cadmium and ruthenium. On the whole the manuscript is written reasonably clearly. However, some sentences are incomplete or muddled and there are a number of typographical and other errors.

The lack of any X-ray characterized structures in the domain of giant multi nuclear complexes is a problem for demonstrating that the claimed architecture has been obtained. The poor quality and complex NMR spectra do not help either, but the authors manage to support their results with various techniques which are ultimately convincing.

The following points need addressing:

The choice of the name 'Star of David' and, in particular, 'Star of Texas' for these structures is not, in my opinion, particularly appropriate nor useful. Star of David has some justification as the molecular structure can reasonably be seen as a 2D representation of the Star of David. However the Star of Texas (or Lone Star) is a particular type of five-pointed star possessing either 3D-bevelling or no internal lines, and so has a different topology to the synthesized molecule, the only similarity being that they are (different types of) five-pointed stars. Maybe the authors should simply call their structures 'supramolecular hexagram and pentagram' which is exactly what they are.

Response to this point: We greatly thank you for the professional and extensive suggestions to our manuscript. After reading the background knowledge about the Star of David, just for purposes of comparison and consistence, we name the five-pointed star as Star of Texas. In fact, this pentagram structure is a little different from named exact "Star of Texas". In our first opinion, we intend to describe their general shape and make it easy to comparison, then following the reference with similar structures, we chose the name. Also, using Star of Texas may generate broader impacts in non-scientific community, and ultimately attract more students for STEM majors. Therefore, we really appreciate it if we can use Star of Texas to convey these molecules.

2. In line 61 hexameric should be tetrameric.

Response: Thanks for your careful check; we have revised it following your suggestion.

3. In lines 75-76 the authors state that "To date, few one-step, Star of David molecular assemblies of (?) have been reported in reasonable yields due, in part, to its large hollow, non-compact scaffold" but no references are included to

support this. Alexandropoulos et al.'s work (Chem. Comm. 2016, 52, 1693) should be referenced and the authors' rationale as to why these assemblies are rarely reported is just unsupported, somewhat random, speculation.

Response to this point: Thanks for your suggestion about the newly reported paper with the topic of Star of David; the reference was added to the reference 20 now. Our first draft was finished in the end of 2015. That's the reason we did not include Alexandropoulos' work in 2016 at that time.

As far as we know, the published assemblies of the Star of David are mainly focused on link structure, which is not exactly 2D planar Star of David structure.

These terpyridine-based hexagram and pentagram involve two ligands (V and X), direct assemblies are impossible to obtain discrete supramolecules. In our study, these metallo-ligands can be adjusted according preliminary geometrical analysis of the target molecules by rational design.

4. In line 104 the authors suggest that Zn²⁺ could be used in the self-assembly reactions but there is no further mention of this. It would be better not to mention it at all if no experiments were actually performed using this metal.

Thanks, you are right; the Zn²⁺ part has been removed from the text.

5. Throughout the manuscript and ESI, metal sources are represented as Mⁿ⁺. It would be clearer to specify the full metal salt used, e.g. FeSO₄·7H₂O, Cd(NO₃)₂·6H₂O, especially in schemes. The counter-ion for charged species is not stated in their assignment section, e.g. the 'Five star of Star-of-Texas' is referred to as [Fe₁₀(LA)₅]³⁰⁺ in section 2.8 of the ESI. Similarly, counter ions are not stated in figures of the NMR spectra for charged species, e.g. in figure S41 the species is stated to be [Fe₁₀(LA)₅]³⁰⁺. As counter ions can influence NMR spectra of charged species they should be stated.

Thanks you for reminding us the details. In the process of the assembly, actually there are a mixtures of several kinds of anions, for example NO₃⁻, Cl⁻, PF₆⁻, SO₄²⁻, depending on the ligands and metal salt used, but it will be eventually exchange to PF₆⁻ using excessive NH₄PF₆ and precipitate from the solvents. For convenience, we omitted all anion. According to your valuable suggestion, we added the anions in the manuscript to make the manuscript more tractable.

6. The complex notation "<tpy-Mn+-tpy>" could be represented in a neater way. Perhaps simply [M(tpy)₂]ⁿ⁺ or [M(tpy)(tpy')]ⁿ⁺.

Thanks for the suggestion! "<tpy-Mn+-tpy>" could emphasize the bridging between two different terpyridine ligands through coordination. Using [M(tpy)₂]ⁿ⁺ or [M(tpy)(tpy')]ⁿ⁺ may imply the complexed ligands are identical ones. Meanwhile, the Mⁿ⁺ could coordinate with only one ligand to form mono-coordinating complex adduct.

7. In the Results and Discussion section (line 98) the authors claim that the [M₃V₃] complexes are "enthalpically favoured" but offer no evidence that this is

the case. Surely there is no enthalpic advantage to forming this triangle over the hexa- or nona-metallic species, only an entropic advantage.

Yes, you are right, the entropic advantage is the main reason here. The sentence has been modified.

8. In lines 169-171, the authors compare the potential stabilities of self-assembled constructs to five- and six-membered rings. This comparison appears tenuous and a little confusing and it does not have any relevance to the observations.

Response: The original idea of designing metallo-ligand LA is for the assembly of Star of David, but we obtained the Star of Texas instead. So we speculate that the different side length and the coordination features of metal ions may play a key role. Surely, forming pentagram instead of hexagram by assembling with LA and metal, the complexing geometrical orientation must be adjusted according to the different side length of triangle, which is not a real equilateral triangle.

Ru/Cd Five Star

9. Though the use of computer generated "theoretical" molecular diameters is fine for estimating the sizes of the reported constructs, some more detailed explanation on software/techniques used etc. should be provided in the ESI.

Response: Thanks! The detailed information about theoretical techniques has been added to the ESI.

10. The captions for Figures 4 and 6 should contain solvent/temperature/magnet strength for NMR data.

Response: Sorry for those missed information, which was revised.

11. The caption for Figure 5 says "Synthesis of key metallo-ligand LB and hexagonal supramolecular Star-of-David assemblies" but only the self-assembly steps are shown. The caption needs altering to reflect this.

Response: Thanks for reminding and the description had been revised.

12. The TEM images do not appear to support the given data. For example, Figure 6E shows dots of radius approx. 3.3 nm if the scale bar is correct, rather than the reported 7.3 nm. There is a similar discrepancy in Figure 6F.

Response: Sorry for not carefully illustrating the radius, the modified TEM images had been added to replace the original ones. Actually, the dots should only present the radius of patterned 30 and 36 metal ions of Stars of Texas David, respectively; the organic parts may not clearly show in the TEM. Therefore, the dots showing in TEM images could be smaller than the calculated molecular radius.

13. In the ESI, scheme S2 is not correct: the methoxy groups are in ortho and meta and not para and meta. Molecule S2 should be on the arrow for synthesis of LA from L-Br.

Response: All the errors had been corrected.

14. The procedure for synthesis of complex S4 does not have a yield.

Response: The yield had been checked and added to the procedure.

15. The procedure for synthesis of complex L-Br states that 150.6 g rather than 150.6 mg of X was used.

Response: The mistake has been corrected.

16. All figures of NMR spectra in the ESI should include the solvent used in the caption.

Response: We had added the solvents to the caption.

17. The ^1H NMR spectrum in figure S41 has a high signal: noise ratio. Hence the integrations shown do not match well with those stated for the assignment of $[\text{Fe}_{10}(\text{LA})_5]^{30+}$. This is also the case for $[\text{Fe}_{12}\text{V}_3(\text{LB})_3]^{36+}$.

Response: You are right. Generally, ^1H NMR of terpyridine-based Fe^{2+} complex frequently illustrates a broad signal or noise baseline, probably due to the trace amount of oxidative Fe^{3+} contamination.

We modified the description about the integration parts of ^1H NMR, thanks for the suggestion.

18. Some NMR spectra are reported to have been recorded operating at a ^1H resonance frequency of 600MHz (e.g. Figure S8). This spectrometer is not recorded in the General Procedures section.

Response: The 600 MHz NMR spectrum measured in Prof. Li's lab, we are sorry for the missing, now the information had been added.

19. In section 2.9 the ESI-MS peak at $m/z = 942.4$ is incorrectly assigned as the 19+ rather than the 20+ species.

Response: The problem was fixed.

20. The ^1H NMR spectrum (figure S44) of $[\text{Fe}_{12}\text{V}_3(\text{LB})_3]^{36+}$ has not been assigned (section 2.10). The quality of the spectrum is similar to figure S41, which has been assigned (section 2.8).

Response: We have revised it according to the NOESY and ^1H NMR spectrum.

21. The caption of figure S44 is highlighted and the species should read $[\text{Fe}_{12}\text{V}_3(\text{LB})_3]^{36+}$ instead of $[\text{Fe}_{12}\text{V}_3(\text{LB})_3]^{36+}$.

Response: We have corrected it.

Reviewer #2 (Remarks to the Author):

In supramolecular chemistry arena, the construction of Star-of-Texas and Star-of-David has proven to be a fascinating challenge. In this manuscript, through a stepwise synthetic strategy and self-assembly approach, Li and Wang et al. carefully designed and prepared two new pentagonal and hexagonal metallosupramolecules possessing the Star-of-Texas and Star-of-David motifs. The Star-of-Texas and Star-of-David architectures were fully characterized by high resolution ESI-MS, TWIN-MS, NMR, 2D-NOESY, and DOSY spectroscopies as well TEM. This is an interesting work that I would recommend to be published in Nature Communication after minor revision.

1. During the past years, Lehn and Leigh groups have also successfully constructed a series of Star-of-Texas and Star-of-David architectures through the coordination-driven self-assembly (J. Am. Chem. Soc., 1997, 119, 10956; Nat Chem., 2014, 6, 978-982; Nat. Chem., 2012, 4, 15-20; etc). The authors need to discuss the advantages of their method compared to the methods reported by Lehn and Leigh to prepare Star-of-Texas and Star-of-David architectures;

*Response: Thanks for Referee's professional reminding about the advantage of our strategy compared to Lehn and Leigh's Star-shaped architectures. Self-assembly of discrete Stars of David or Texas is challenging. Lehn and Leigh groups had constructed them based on **pure organic bipyridines** with Fe^{2+} to form **helicates** or **link-like catenanes structures**; in fact, they are not exactly planar Stars of David or Texas. Particularly, the catenanes in Leigh's study required post-assembly ring-closing olefin metathesis of the adjacent terminal bipyridine residues. Hererin, we use the **metallo-terpyridine ligands**, as the building blocks to assemble with metal ions to form a **rigid planar** Stars of David or Texas. For the first time, we were able to design and assemble giant and complicated supramolecular architecture using terpyridine-based supramolecular chemistry.*

We are modified the introduction to emphasis the differences of the published star-shaped architectures and our current researches.

2. For preparing the Star-of-Texas and Star-of-David architectures, $\text{Cd}(\text{NO}_3)_2$ and FeCl_2 have been used in this work. I wonder what will happen if CdCl_2 and $\text{Fe}(\text{NO}_3)_2$ are used.

Response: Both are working. NO_3^- and Cl^- gave similar result according to our study. After the assembly, all the anion would be replaced by PF_6^- through precipitation. The precipitates will be effectively washed with D. I. water and MeOH to remove the excess of PF_6^- .

3. In order to prepare the Star-of-Texas architecture via one-pot procedure, ligand V and ligand X were mixed directly with FeSO_4 , Unfortunately, a mixture of small triangle, bowtie-like double-triangle, and triple-triangle along with a small amount of unidentified metallo-polymer was formed. Considering the successful preparation of Star-of-Texas architecture by using FeCl_2 , I also wonder what will happen if FeCl_2 is used for preparing the Star-of-Texas architecture via one-pot procedure.

Response: Yes, we have used FeCl_2 to perform this experiment you mentioned and got the similar results. Because the Zn^{2+} and Cd^{2+} have mild reaction condition and easy to obtain the most thermodynamic stable products, we typically use with Zn^{2+} and Cd^{2+} as our first choice.

4. In page 7, the authors described that "These observed chemical shifts and UV-vis measurements (see ESI, Figure S54) indicated the expected <tpy- Fe^{2+} -tpy> linkages were formed". The detailed discussion of the UV-vis results should be provided. Especially the relationship between the UV-vis spectra and the architecture structure should be discussed in detail.

*Response: The tpy- Fe^{2+} -tpy has a typical absorption at ca. 570nm and tpy- Ru^{2+} -tpy at ca. 490nm, which was reported by many terpyridyl literatures (for example: Chem. Eur. J. **2004**, 10, 1493). The UV-vis description had been added to the manuscript.*

5. In the past few years, the author's group has prepared a series of metallosupramolecular architectures with fascinating structure based on terpyridine (tpy) skeleton (for example, J. Am. Chem. Soc., 2014, 136, 6664–6671; J. Am. Chem. Soc., 2014, 136, 10499–10507; J. Am. Chem. Soc., 2015, 137, 1556–1564; Chem. Commun., 2016, 52, 9773; etc.). Although the construction of such beautiful metallosupramolecular architectures is interesting, it would be better if some applications will be explored for these architectures.

*Response: We have tried to find their applications, for example, in PDT therapy, light catalyzing decomposing H₂O, gel formation, OTFT device and so on. In fact, Prof. Li's group has made some progress (Nano Lett. **2015**, 15(9), 6276) in the field of self-healing materials. At this stage, the stepwise strategy gives us more chance to assemble more sophisticated architectures, along with the improvement of the ligands synthesis and structure designation, the application will be accelerated. Thanks for this valuable suggestion.*

6. Some errors should be corrected, for instance, ml should be corrected as mL; a space should be added between the value and the unit.

Response: We fixed the errors.

Reviewer #3 (Remarks to the Author):

The article by Jiang et al is a very thorough piece of work. The authors report the systematic synthesis of coordination assemblies based on Star-of-Texas and Star-of-David arrangements. The work is undoubtedly clever and the strategy interesting but I am left wondering whether it is a significant advance beyond previous work?

First, we appreciate Reviewer 3's professional comments on this manuscript. The design and self-assembly of our Stars of David and Texas was based on the metallo-ligands. The supramolecular Stars of David and Texas are planar and rigid with all the building blocks in the same planes; second, the final assembly are high efficient, especially with Cd²⁺ and Fe²⁺, the purification processes are also simple; third, this strategies will give a guide on how to design a higher molecule-weight single molecular architecture.

The authors identify a large number of previous papers on related materials and even acknowledge the previous study of a Star-of-David catenane (ref 20) which in a similar fashion to the current study builds upon the circular helicates reported Lehn twenty years ago (references 2 and 3). Is this current study a significant conceptual advance on these previous studies? Ultimately I don't think so and therefore I do not see that the paper is sufficiently novel for publication in Nature Communications.

Response: Lehn's structures the Referee mentioned are helicates instead of a planar platform. From geometrical and chemistry point of view, hexameric circular helicate structures are more stable and easier to survive in a solvent condition, since the connection between the ligands and metals could be hold through the helicate twining round. The supramolecules presented in this

manuscript are totally 2D planar structures. The whole molecular connections are in a plane, which are less stable, so the applied metallo-ligand needs to be carefully designed. Of course, the helicates and catenane look similar, but the composite units and the way of their construction are definitely different. In this work, we want to introduce a totally new story of design and self-assembly of a novel series molecules which are not succeed before.

Bearing in mind my reservations about the novelty of this work I do have some questions about the work as reported. Lehn's original work on circular helicates identified the role of the anion as significant in determining the nature of the structure of the resulting coordination assembly. What is the role of the anion in the current study? Is the size and shape of the PF₆⁻ anion significant in determining the target Star-of-Texas or Star-of-David arrangements? Can the balance between different arrangements be modified by different anions. Additionally is the anion trapped within the structure as is the case for Lehn's circular helicates? Have the authors recorded ¹⁹F NMR which would help to identify the role of the anion?

The authors need to seriously consider these additional studies and clarify if there are additional factors which control the assembly of their structures.

*Response: Thanks for very thoughtful point of anion roles in the assembly. Profs. Lehn and Leigh both found that the anions, such as Cl⁻, could be as a significant role during the coordination assembly. Notably, Lehn and Leigh's supramolecules are mostly the helicates or catenane structures, the whole molecules (not in a plane) could be self-adjusted by the possible environmental perturbation, such as anion (J. Am. Chem. Soc. **2015**, 137, 9812); also Lehn and Leigh's molecules and molecular central holes are smaller, which are less than 1 nm. In their cases, the anion, such as Cl⁻, is suitable to fit the holes. In our investigation, we have monitored the molecular Stars of David and Texas in different anions, such as PF₆⁻, Cl⁻, NO₃⁻, BF₄⁻ and OTf⁻, by ESI-MS measurements, we found there was no molecular structure change during the different anions use. This may be attributed to the significantly large central holes (more than 3.5 nm) and molecular-weight (more than 20,000 Da). In our case, the metal ions gave most impact on the final assembly. According to the published articles and the experience in our lab, the Zn²⁺ and Cd²⁺ have more stronger self-correction abilities to form stable discrete architectures; however, Fe²⁺ are liable to form disorder polymer in ambient and order-assemblies at high temperature and Ru²⁺ do not tend to form orderly architectures with less selectivity.*

*A few publications did report that concentration of the terpyridine-based coordinated complexes is an important factor for determining the final products based on entropic effects (For example, J. Am. Chem. Soc. **2014**, 136, 18149). In this investigation, at very high concentration of supramolecular Star of Texas ([Cd₁₀LA₅]³⁰⁺), a very small amount of molecular Star of David ([Cd₁₂LA₆]³⁶⁺) was only detected by the ESI-MS spectrum, but the major product was still the*

Star of Texas. Nevertheless, the structure of Star of Texas ($[\text{Fe}_{10}\text{LA}_5]^{30+}$) kept stable in the different concentrations.

Related to the influence of anion, PF_6^- is a natural choice to precipitate the assembled complexes from the solvents. Due to a certain amount of solubility, other anions are seldom used. We ran the ^{19}F NMR as you suggested and found that the structures with PF_6^- as the anion kept steady. Fe^{2+} generated Stars were precipitated out from ethylene glycol/diethyl ether, in which the NH_4PF_6 can be completely washed off.

Reviewer #4 (Remarks to the Author):

This is a potentially interesting ms. However, the write up is very confusing, cumbersome and difficult to follow. There are also numerous English problems. I doubt that coauthor Newkome saw this ms and if he did he should never have let it be submitted with such poor English and write up.

Response: We are deeply sorry for the English writing. The English and grammar are carefully polished. We apologies again for this inconvenience.

However, there are even more serious issues. Mass spec does not prove structure at best it can only give stoichiometry. That leaves just proton NMR for characterization and frankly that is simply not sufficient for complex molecules like these. Just as an example the H-NMR of both compounds are essentially the same and makes it impossible to really assign structure. Hence until there is

better and more definitive structure proof and a much better and clear write up this ms is NOT ready for publication in any journal let alone a Nature Journal.

*Response: About this question, we do find many references about pyridine-based supramolecular assemblies only used ESI-MS to determine the giant structures, for example, J. Am. Chem. Soc. **2011**, 133, 11967–11976, J. Am. Chem. Soc. **2014**, 136, 18149–18155, PNAS, **2015**, 112, 18, 5601. J. Am. Chem. Soc. **2014**, 136, 10499–10507. J. Am. Chem. Soc. **2014**, 136, 4460–4463. J. Am. Chem. Soc. **2014**, 136, 5811–5814. In assembly chemistry, single crystal X-ray diffraction is a most powerful evidence to confirm an exact 2D or 3D structures. Unfortunately, we have tried many times, our giant supramolecules are difficult to obtain a single crystal structure, especially the 2D motif (Angew. Chem. Int. Ed. **2015**, 54, 9224–9229). To identify the assembly structures, Prof. Stang and the other researchers have frequently used the powerful new techniques of HR-MS to prove the assemblies structures. Just like peptide synthesis (J. Am. Chem. Soc. **2011**, 133, 759–761), when the single-crystal data is difficult to obtain, the MS spectrum are the strongest persuasion with the precursor each amino acid participate in the synthesis are unmistakable. In addition to regular ESI-MS, traveling wave ion mobility-mass spectrometry (TWIM-MS) as a powerful method based on collision cross section have been widely used a variety of different supramolecular chemistry systems (J. Am. Chem. Soc. **2012**, 134, 9193; Angew. Chem. Int. Ed. **2011**, 50, 8871; J. Am. Chem. Soc. **2011**, 133, 11967).*

Following your question, except the intend to grow the single crystal, we have also tried many times in order to obtain a clear TEM, AFM or STM images to visually detect the expected pattern of five- or six-shaped stars. We do find the TEM images of six-star shaped Star of David can be identical to a cycle pattern, which may give an additional evidence for a cyclic structure that indicated the corrected connections between the metallo-ligand and metal. Unfortunately, the TEM image of molecular Star of Texas has still been imaged as the round dots. With the help of Prof. Xu at University of Georgia, we obtained more solid evidence by AFM and STM images which were added to the main text as the Figures 7 and the supporting information as the Figure S56. We really appreciate it if you can reconsider our manuscript based on our efforts.

TEM images of Star of David.

A and B: AFM images of Star of David; C and D: STM images of Star of David.

A and B: AFM images of Star of Texas; C and D: STM images of Star of Texas.

Reviewers' comments:

Reviewer #1 (Remarks to the Author):

Although the authors have addressed most of my concerns satisfactorily, there are still some aspects of the manuscript that are unsatisfactory:

1. I explained to the authors that the term 'star of Texas' is inappropriate because the structure is not the same as the star of Texas, and their response is that they realize it's not the same structure but they want to use the name anyway. That is not a reasonable response.
2. At lines 166-167 and 232-233, the authors are saying that the calculated hydrodynamic radius agrees well with the molecular modelling but without giving any numbers. As this is one of the important proofs for supporting their claims, they should more detail.
3. At lines 234-240, the TEM results are still not convincing especially in regard of the scale. The dots (2-3 nm) are certainly not the size of the expected complexes and not even close to the metal ion "crowns" as presented in the answer to my point 12 (5.2 nm and 6.5 nm).
4. For the new addition of the AFM images, they are convincing for the six-star (figure 7) but NOT AT ALL for the five-star (figure S56). In this figure, according to the scale between pictures B and D, the size of the five-star changes from 6 nm to 28 nm. There are no clear signs of the presence of the discrete five-star provided by the AFM experiments.

Reviewer #2 (Remarks to the Author):

The authors have made substantial efforts to address the comments of the reviewers. In most instances, they have successfully addressed the comments raised. From my point of view two major points remain.

The authors do not appear to have modified the main text or supplementary information to comment on the effect, or lack of it, of anions upon the structure that is formed. This is a significant point and distinguishes the strategy reported in this manuscript from previous studies (notably by Lehn and Leigh). The authors must address this point in the text.

Secondly I am still unconvinced by the novelty of the approach. Yes, there are differences between the current study and those of Lehn and Leigh. However, are those differences significant? Just because the authors make structures which "are in a plane" does that make their strategy significant? I was left asking the question, so what? Why does it matter that their structures "are in a plane"? I don't think that this point has been addressed and therefore I am still unconvinced by the originality/significance of the manuscript. This has to be an editorial decision.

The authors also need to cite a very recent publication which has been published in the last few weeks - Science, 355, 159-162 (2017).

Response to Referees' comments:

Reviewers' comments:

Reviewer #1 (Remarks to the Author):

Although the authors have addressed most of my concerns satisfactorily, there are still some aspects of the manuscript that are unsatisfactory:

1. I explained to the authors that the term 'star of Texas' is inappropriate because the structure is not the same as the star of Texas, and their response is that they realize it's not the same structure but they want to use the name anyway. That is not a reasonable response.

Response to this point: We greatly appreciate your professional and extensive suggestions to our revised manuscript.

Indeed, the five-point pentagram structure is different from pre-named "Star of Texas". We have change the Title to 'Supramolecular Hexagram and Pentagram', and also modified the manuscript.

2. At lines 166-167 and 232-233, the authors are saying that the calculated hydrodynamic radius agrees well with the molecular modelling but without giving any numbers. As this is one of the important proofs for supporting their claims, they should more detail.

Response to this point: The calculated hydrodynamic radius had been added, the calculated r_H values from DOSY measurements are the average of inter- and outline of molecules, which are slightly smaller than the computer generated molecular radius.

3. At lines 234-240, the TEM results are still not convincing especially in regard of the scale. The dots (2-3 nm) are certainly not the size of the expected complexes and not even close to the metal ion "crowns" as presented in the answer to my point 12 (5.2 nm and 6.5 nm).

Response to this point: We inquired the TEM testing center in Beijing, a outsource Company for TEM tests, about this size issue; their response is probably due to the calibration of scale bar, especially when imaging in a higher magnification.

At this point, since the AFM and TEM images of pentagram and hexagram had clearly demonstrated the molecular morphologies, we are going to delete the TEM images in Figure 6 and also the corresponding Text.

4. For the new addition of the AFM images, they are convincing for the six-star (figure 7) but NOT AT ALL for the five-star (figure S56). In this figure, according to the scale between pictures B and D, the

size of the five-star changes from 6 nm to 28 nm. There are no clear signs of the presence of the discrete five-star provided by the AFM experiments.

Response to this point: We appreciate very much the reviewer's concerns on the AFM images of the five-star (Figure S56). This is really about the technique issue of AFM imaging. The resolution of AFM imaging depends on not only the sample but also the "Mode" of imaging and the sharpness of the AFM tip. The AFM tips we used have a diameter of 2-10 nm (depends on which individual tip, which cannot be determined beforehand). The tip diameter effect of the resolution is called the imaging "broadening" effect, which means the feature size is broadened. Recent research found that even if the carbon nanotube AFM tip (the sharpest tip one can get up to now) is used, the width of the imaged feature can be doubled (Nanotechnology 23 (2012) 405705). In order to preserve (not damage) the molecule structure, in this study we used a TopMAC mode AFM where the cantilever needs to be coated with a magnetic material. Doing so will further increase the tip diameter so the large scale of the AFM image is simply the results of tip broadening effect.

Nevertheless, we test a few more tips and now the AFM images are very convincing. As is shown in the revised Figure S56, the minimized "broadening" due to an extremely sharp tip enabled the acquired image to have the size (~12 nm) close to the STM image (7-8 nm). The 5-star structure is clearly shown in the amplitude image.

Figure S56

Reviewer #2 (Remarks to the Author):

The authors have made substantial efforts to address the comments of the reviewers. In most instances, they have successfully addressed the comments raised. From my point of view two major points remain.

The authors do not appear to have modified the main text or supplementary information to comment on the effect, or lack of it, of anions upon the structure that is formed. This is a significant point and distinguishes the strategy reported in this manuscript from previous studies (notably by

Lehn and Leigh). The authors must address this point in the text.

Response to this point: Description of different anion as counterions upon the structure has been added to the Text (line 240), a supporting figure has been added to the Electronic Supporting Information (Figure S57). Similar results were also applied to hexagram, no structure change was detected. Thanks for the suggestion!

Figure 2: ESI-MS illustration of exchanging BF_4^- from NO_3^- , no structure change was detected.

Figure 3: ESI-MS demonstration of exchanging PF_6^- from NO_3^- , ClO_4^- from PF_6^- , and the mixed anions, no structure change was detected.

Secondly I am still unconvinced by the novelty of the approach. Yes, there are differences between the current study and those of Lehn and Leigh. However, are those differences significant? Just because the authors make structures which “are in a plane” does that make their strategy significant? I was left asking the question, so what? Why does it matter that their structures “are in a plane”? I don’t think that this point has been addressed and therefore I am still unconvinced by the originality/significance of the manuscript. This has to be an editorial decision.

Response to this point: Thanks a lot for your extensive point related to the novelty of this manuscript! In the study by Profs. Lehn and Leigh, the assembled structures were determined by metal ions and counterions. For instance, 1D triple helix (R. Krämer, J.-M. Lehn, A. D. Cian, J. Fischer, *Angew. Chem. Int. Ed. Engl.* **1993**, 32, 703) was assembled instead of 2D penta- and hexameric cyclic helicates assembled by Fe(II). In our case, the self-assembly was precisely controlled as either penta- or hexagram through metal-organic ligand design instead of counterions and metal ions. Therefore, all previous study based on bipyridine of penta- and hexagram by Profs. Lehn and Leigh was mainly concentrated on Fe(II) with Cl⁻ or sulfate. Without structural variation, our strategy based on terpyridine will allow us to introduce a variety of metal ions with weak coordination in the combination with Ru(II), e.g., Fe(II), Zn(II), Co(II), Ni(II) and Cd(II). Similarly, Ir(III) and Os(II) might also be employed to replace Ru(II) because of their strong coordination (J.-P. Sauvage, J.-P. Collin, J.-C. Chambron, S. Guillerez, C. Coudret, V. Balzani, F. Barigelletti, L. D. Cola and L. Flamigni, *Chem. Rev.*, **1994**, 94, 993). Considering that these terpyridine-metal complexes have demonstrated a myriad of applications due to their tunable electrochemical, photo- and electroluminescence properties (Winter, A.; Hager, M. D.; Newkome, G. R.; Schubert, U. S. *Adv. Mater.* **2011**, 23, 5728. Winter, A.; Hoepfener, S.; Newkome, G. R.; Schubert, U. S. *Adv. Mater.* **2011**, 23, 3484.), this penta- and hexagram system may serve as a robust model system to study light harvesting devices, optical display, switches, and energy storage.

The authors also need to cite a very recent publication which has been published in the last few weeks - *Science*, 355, 159-162 (2017).

Thanks! The paper had been added.